# Nutritive Value of Some Concentrate Feedstuffs for Guinea Pigs (*Cavia porcellus*)

**DOI:** 10.3390/ani14213142

**Published:** 2024-11-01

**Authors:** Gilbert Alagón, Gardenia Tupayachi, Wagner Villacorta, Carla Taco, Moises Jancco, Eloy Zuniga, María del Carmen López-Luján, Luis Ródenas, Vicente Javier Moya, Eugenio Martínez-Paredes, Enrique Blas, Juan José Pascual

**Affiliations:** 1Facultad de Agronomía y Zootecnia, Universidad Nacional de San Antonio Abad del Cusco, Avenida de la Cultura 733, Ap. Postal 921, Cusco 08000, Peru; gardenia.tupayachi@unsaac.edu.pe (G.T.);; 2Instituto de Ciencia y Tecnología Animal, Universitat Politècnica de València, Camino de Vera s/n, 46022 València, Spain; malolu@upvnet.upv.es (M.d.C.L.-L.); eblas@dca.upv.es (E.B.); jupascu@upv.edu.es (J.J.P.)

**Keywords:** guinea pig, nutritive value, corn, barley, wheat bran, soybean meal, pigeon pea, Leucaena leaf meal

## Abstract

Traditionally, guinea pig breeding has been performed in small-scale production units, with forage and domestic kitchen or crop waste, to alleviate food insecurity and enhance family nutrition. Nevertheless, the development of more effective guinea pig production systems is becoming increasingly pertinent as a means of augmenting farmers’ income and fostering rural advancement. To achieve this, it is essential to use improved genetic lines with greater productive potential than the unselected native guinea pig (creole), such as those generated by several Peruvian institutions, in conjunction with balanced feeds that meet their nutrient requirements. To formulate balanced feeds at the lowest possible cost, it is necessary to have the most precise knowledge possible of the nutritive value of the available feedstuffs. However, specific information in this regard is very scarce in the guinea pig case. The current work contributes to the knowledge of the nutritive value of six widely used concentrate feedstuffs for this species (corn, barley, wheat bran, soybean meal, pigeon pea, and Leucaena leaf meal), determining two of the most interesting parameters, their digestible energy content, and the digestibility of their protein.

## 1. Introduction

In general, the guinea pig is considered a laboratory animal. However, guinea pig meat constitutes a component of the human diet in specific geographical areas, notably in the Andean region (Peru, Bolivia, Colombia, and Ecuador). In 2022, the production of guinea pig meat was 13,400 and 5300 t in Peru and Bolivia, respectively, according to FAOSTAT data [1]. To a lesser extent, the consumption of guinea pig meat has also been common in the different countries of sub-Saharan Africa for several decades [2].

Traditionally, guinea pig breeding has been conducted in small-scale production units, with forage and domestic kitchen or crop waste, to alleviate food insecurity and enhance family nutrition [3]. Nevertheless, the development of more effective guinea pig production systems is becoming increasingly pertinent as a means of augmenting farmers’ income and fostering rural advancement. To achieve this, it is essential to use improved genetic lines, with greater productive potential than the unselected native guinea pig (creole), such as those generated by several Peruvian institutions [4], as well as balanced feeds that meet their nutrient requirements.

To formulate balanced feeds at the lowest possible cost, it is necessary to have the most accurate information possible on the nutritive value of the available feedstuffs. However, information in this regard in the guinea pig case is very scarce. Thus, it is common to find articles in which guinea pig diets with different levels of digestible energy (DE) or metabolizable energy (ME) are compared without providing details on how these values were experimentally determined or calculated [5,6,7,8]. To our knowledge, Paredes and Goicochea [9] are the only authors to indicate that the dietary DE contents of guinea pig diets were calculated from the DE contents for rabbits of the different feedstuffs, as provided by de Blas et al. [10].

The most complete information was provided by Castro-Bedriñana and Chirinos-Peinado [11], who determined the apparent total tract digestibility (aTTD) of organic matter (OM), crude protein (CP), ether extract (EE), crude fibre and nitrogen-free extractives for a total of 63 feedstuffs (12 kitchen or agro-industrial waste, 31 forages, three energy concentrates, and 17 protein concentrates). However, neither the number of animals used nor the precision of the values obtained were reported. In their work, the DE content was calculated from the total digestible nutrients (TDNs), according to Crampton et al. [12], and the ME content was determined by multiplying the DE content by the factor 0.82 obtained in steers [13].

The objective of this work is to ascertain the nutritive value, in particular, the DE content and the aTTD of CP, of six commonly used feedstuffs in guinea pig feeds: three energy concentrates (corn, barley, and wheat bran) and three protein concentrates (soybean meal, pigeon pea, and Leucaena leaf meal).

## 2. Materials and Methods

### 2.1. Evaluated Feedstuffs and Experimental Diets

The six feedstuffs evaluated (corn, barley, and wheat bran as energy concentrates; soybean meal, pigeon pea. and Leucaena leaf meal as protein concentrates) were initially subjected to analytical characterization, as shown in Table 1.

Three basal mixtures were formulated: BM-I (based on alfalfa, soybean meal, and barley hulls) to evaluate corn, barley, and wheat bran, BM-II (based on barley, alfalfa, barley hulls, wheat bran, and corn) to evaluate soybean meal, and BM-III (like BM-II but using other batches of feedstuffs) to evaluate pigeon pea and Leucaena leaf meal. The aforementioned basal mixtures were then combined with a mineral–vitamin premix at a ratio of 3.5% to yield three basal diets: basal diet-I, basal diet-II, and basal diet-III. On the other hand, six test diets were obtained by replacing the corresponding basal mixture with 40% corn (corn diet), 50% barley (barley diet), 45% wheat bran (wheat bran diet), 35% soybean meal (soybean meal diet), 30% pigeon pea (pigeon pea diet), or 30% Leucaena leaf meal (Leucanea leaf meal diet), maintaining 3.5% of a mineral–vitamin premix across all diets. The composition of the nine experimental pelleted diets is shown in Table 2.

### 2.2. Digestibility Trials

The digestibility trials were carried out at the Kayra farm at the Faculty of Agronomy and Zootechnics of the National University of San Antonio Abad of Cusco (Perú) (UNSAAC), located in San Jerónimo, Cusco, at an altitude of 3290 m above sea level, in a building equipped with a ventilation and temperature (18–22 °C) electronic control system. The guinea pigs were subjected to a photoperiod comprising 14 h of light and 10 h of darkness.

Sixty-three unsexed guinea pigs of the “Peru” line [4] from the same farm were used. They were between 20 and 25 days old (weaned at 15 days old) and randomly distributed among the nine experimental diets (7 animals/diet). The animals were housed in individual galvanized metal cages, measuring 40 × 25.5 × 25.5 cm, with a metal mesh floor and cover, equipped with hopper-type feeders, with a capacity of approximately 400 g, and nipple-type drinkers. A perforated galvanized metal tray was placed under the cages, enabling the retention of faeces and the prompt evacuation of urine. The animals were provided with feed and water ad libitum. A proper ethical evaluation of the trials was not performed (because of the lack of an ethics committee for animal experimentation in the UNSAAC). Nevertheless, the experimental protocols were drawn up by the team from Universitat Politècnica de València (Spain) (UPV) in accordance with Spanish legislation on this matter (Royal Decree 53/2013) [14].

The adaptation period to the experimental diet and to the individual cage lasted for 10 days, after which the feed intake and faeces excretion were monitored for a further 5 days. The faeces of each animal were collected daily at the same time, placed in tightly closed and labelled plastic bags, and then frozen until they were dehydrated, weighed, ground, and sampled for analysis in the laboratory.

### 2.3. Chemical Analyses

During the preparation of the experimental diets, the dry matter (DM) content of the basal mixtures, test feedstuffs, and mineral–vitamin premixes was determined in the laboratory at the Faculty of Agronomy and Zootechnics of the National University of San Antonio Abad of Cusco. Subsequently, in the animal feeding laboratory at the Institute for Animal Science and Technology of the UPV, the test feedstuffs were analysed for DM, ash, gross energy (GE), CP, EE, starch, neutral detergent fibre (aNDFom), acid detergent fibre (ADFom), lignin (sa), and amino acids, while the experimental diets and individual faeces were analysed for DM, ash, GE, CP, aNDFom, ADFom, and lignin (sa).

Methods of the AOAC [15] were used for DM (934.01), ash (942.05), CP (990.03, the Dumas method, CN628 Elemental Analyzer, LECO, St. Joseph, MI, USA) and EE (920.39, with acid-hydrolysis of the samples prior to extraction). The starch content was determined according to Batey [16], by a two-step enzymatic procedure with solubilization and hydrolysis to maltodextrins with thermostable α-amylase followed by complete hydrolysis with amyloglucosidase (both enzymes from Sigma-Aldrich, Steinheim, Germany). The resulting glucose was then measured by the hexokinase/glucose-6 phosphate dehydrogenase/NADP system (R-Biopharm, Darmstadt, Germany). In the case of the pigeon pea and pigeon pea diet, a preliminary treatment was conducted with a 1.7M sodium hydroxide solution to also include resistant starch, as described by McCleary et al. [17]. The aNDFom, ADFom, and lignin (sa) fractions were analysed sequentially according to Mertens et al. [18], AOAC procedure 973.18 [15], and Robertson and Van Soest [19], respectively, with a thermostable α-amylase pre-treatment and expressed exclusive of residual ash, by using a nylon filter bag system (Ankom, Macedon, NY, USA). In the case of pigeon pea, the pigeon pea diet, and the corresponding faeces, aNDFom was corrected by subtracting the starch content of their NDF residues analysed with pre-treatment with a 1.7 M sodium hydroxide solution. The amino acid content in the diets was determined after acid hydrolysis with HCl 6N at 110 °C for 23 h, as previously described Bosch et al. [20], using a Waters (Milford, MA, USA) HPLC system comprising two pumps (Mod. 515, Waters), an autosampler (Mod. 717, Waters), a fluorescence detector (Mod. 474, Waters), and a temperature control module. Aminobutyric acid was added as an internal standard after hydrolysation. The amino acids were derivatized with AQC (6-aminoquinolyl-N-hydroxysuccinimidyl carbamate) and separated with a C-18 reverse-phase column Waters Acc. Tag (150 mm × 3.9 mm). Methionine and cystine were determined separately as methionine sulphone and cysteic acid, respectively, after performic acid oxidation and subsequent acid hydrolysis. The GE content was determined by adiabatic bomb calorimetry (Gallenkamp Autobomb, Loughborough, UK).

### 2.4. Calculations and Statistical Processing of Data

The calculations were carried out according to Villamide et al. [21], for the scenario of replacing a basal mixture with the test feedstuff and including a fixed level of mineral-vitamin premix into all experimental diets. Briefly, the inclusion levels of the basal mixture, the test feedstuff, and the mineral–vitamin premix were adjusted based on their DM contents, with the objective of calculating the corrected substitution level (cSL) of the basal mixture for the test feedstuff, excluding the mineral–vitamin premix. On the other hand, it was assumed that the mineral–vitamin premix did not provide OM nor, consequently, GE, CP, aNDFom, or ADFom to the experimental diets. Therefore, the respective digestible contents of the experimental diets were also corrected to consider that they came only from the basal mixture and the test feedstuff. To calculate the digestible nutrient contents of the test feedstuffs, the principle of additivity was assumed, and the following equation was employed:(1)$${DNC}_{TF}=\frac{{cDNC}_{TD}-\left(1-cSL\right)\times {cDNC}_{BD}}{cSL}$$
where DNC_TF_ is the digestible nutrient content of the test feedstuff, whereas cDNC_TD_ and cDNC_BD_ are the corrected digestible nutrient content of the test diet and the basal diet, respectively. cSL is the corrected substitution level of the test feedstuff. Finally, the standard error of DNC_TF_ was calculated with the following equation:(2)$${SE}_{TF}=\frac{1}{cSL}\sqrt{\frac{{cSD}_{TD}^{2}}{n_{TD}}+{\left(1-cSL\right)}^{2}\times \frac{{cSD}_{BD}^{2}}{n_{BD}}}$$
where SE_TF_ is the standard error of DNC_TF_, cSD_TD_ and n_TD_ are the standard deviation and the number of data of cDNC_TD_, respectively, and cSD_BD_ and n_BD_ are the standard deviation and the number of data of cDNC_BD_, respectively.

## 3. Results

Table 3 shows the discrepancy between the analytical composition of the test diets and their calculated composition from the analyses of the corresponding basal diets and test feedstuffs, expressed as % of the calculated composition. The observed deviation was low for GE (ranging from −0.6 to 1.8%) and OM (from 0.0 to 1.0%), intermediate for CP (from −1.4 to 6.7%), and high for aNDFom (from −7.6 to 4.2%) and ADFom (from −6.7 to 5.9%).

Table 4 shows the cSL and the contribution of the test feedstuffs to the analytical composition of their respective test diets calculated from their composition and inclusion level corrected by DM (before excluding the mineral–vitamin premix), which expresses the substitution level for each specific nutrient. The lowest and highest values corresponded to the contribution of corn to the ADFom content of the corn diet (8.4%) and the contribution of soybean meal to the crude protein content of the soybean meal diet (65.1%), respectively.

The aTTD of the experimental diets is presented in Table 5. The feed intake of all experimental diets averaged 49.8 g DM/day, and no health issues were detected in the animals during the digestibility trials.

In general, compared with those of the respective basal diet, the overall apparent total tract digestibility values (aTTD of DM, GE, and OM) of the test diets varied consistently with the main changes induced in their composition by the inclusion of the test feedstuff. Thus, they increased with the inclusion of 40% corn (+7.5 percentage points on average) and 50% barley (+6.3 percentage points on average), given that both feedstuffs are rich in starch and poor in fibre. Conversely, they decreased with the inclusion of 45% wheat bran (−2.6 percentage points on average), which is much poorer in starch and richer in fibre than cereals. Similarly, they increased with the inclusion of 35% soybean meal (+4.0 percentage points on average), which is rich in protein and poor in fibre. Furthermore, they also increased, although to a lesser extent (+1.0 percentage points on average), with the inclusion of 30% pigeon pea, which is less rich in protein and richer in fibre than soybean meal, but they decreased (−2.2 percentage points on average) with the inclusion of 30% Leucaena leaf meal, which is much poorer in starch and richer in fibre than pigeon pea.

The variability in the overall apparent total tract digestibility of the experimental diets was low (coefficients of variation of 2.2, 2.3, and 1.9% on average, for the aTTD of DM, GE, and MO, respectively), slightly higher for the aTTD of CP (coefficient of variation of 3.1% on average), and particularly high for the aTTD of fibre (coefficients of variation of 7.3 and 10.3% on average, for the aTTD of aNDFom and ADFom, respectively).

The DE content and the aTTD of the test feedstuffs, along with their respective standard errors, are presented in Table 6.

Logically, the aTTD of GE and OM of the test feedstuffs varied according to the same pattern, so the arrangement of the test feedstuffs sorted by decreasing values was corn, soybean meal, barley, pigeon pea, wheat bran, and Leucanea leaf meal. The DE content of corn and soybean meal was very similar, while that of barley, pigeon pea, and Leucaena leaf meal was 10, 19, and 23% lower, respectively. Although the lowest aTTD of GE was observed in the Leucaena leaf meal, its DE content was 2% higher than that of wheat bran, attributable to its elevated GE content.

The aTTD of CP was maximum in soybean meal, lower in wheat bran and corn, even lower in barley and Leucaena leaf meal, and minimum in pigeon pea. In comparison with soybean meal, the digestible protein content was markedly lower in the remaining test feedstuffs (56, 67, 68, 85, and 86% in Leucaena leaf meal, wheat bran, pigeon pea, barley, and corn, respectively).

Regarding the aTTD of aNDFom and ADFom, it should be noted that their residual variability was much greater than in the case of the aTTD of GE, MO, and CP (the coefficient of variation of these last averaging 6.6, 5.6, and 11.0%, respectively). The coefficient of variation for the aTTD of aFNDom averaged 27.9% (minimum 20.1% for pigeon pea; maximum 43.8% for corn), and for the aTTD of ADFom averaged 133% (minimum 31.8% for pigeon pea; maximum 301% for corn).

## 4. Discussion

The direct determination of the digestible nutrient content of a feedstuff, by carrying out a digestibility trial using the feedstuff as the unique ingredient of the feed, is only applicable to a limited number of cases, namely, those instances where the feedstuff is relatively balanced, adequately ingested, and does not impair digestive function [21]. For the remainder of the feedstuffs, an alternative method is the substitution approach, which involves the use of a test diet in which an evaluated basal diet is partially replaced by the feedstuff to be evaluated (through calculations based on additivity). In this case, it is essential to plan both the basal diet and the substitution level to ensure that the basal diet and the test diet are not excessively unbalanced. A substitution level greater than 20% is recommended to estimate the DE content (and, logically, other overall digestible nutrient contents, such as the digestible OM content) with good precision, as the differences between both diets are attributed to the test feedstuff [22]. Consequently, the precision of the estimate increases with the substitution level, and it has been suggested that high substitution levels should be employed, even if this results in unbalanced diets [23].

In the current study, the DE, digestible protein, and ADFom contents of the experimental diets ranged between 2970 and 3606 kcal/kg DM, 9.0 and 24.5%, and 10.1 and 19.0%, respectively. The recorded feed intake (49.8 g DM/day on average) can be considered adequate, given that it was 12% higher than that reported in animals of the same age fed only with pelleted diets [9]. Furthermore, no adverse effects on digestive health were observed with any of the experimental diets.

The cSL ranged between 32.1% and 51.6%, and the contribution of the test feedstuffs to the GE and OM contents of the respective test diets showed logically very similar values, which allowed for the estimation of the DE content and the aTTD of OM for all the test feedstuffs with suitable precision (with somewhat lower precision for wheat bran, in which the coefficients of variation were 10.0 and 9.0%, respectively, in contrast to values that averaged 5.9 and 4.9%, respectively, for the rest of the feedstuffs).

The contribution of the test feedstuffs to the CP content of their respective test diets was very high in the case of soybean meal (65.1%), high in the case of Leucaena leaf meal and pigeon pea (on average 48.9%), moderate in the case of wheat bran (40.5%), and low for cereals (on average 22.9%). Consequently, the precision in estimates of the aTTD of CP varied inversely, with the highest level of precision observed for soybean meal (the coefficient of variation was 2.9%), followed by Leucaena leaf meal and pigeon pea (the coefficient of variation was 7.6% on average), then wheat bran (the coefficient of variation was 12.5%), and finally, cereals (the coefficient of variation was 17.5% on average). These findings are comparable to those obtained in rabbits, wherein the variability in the aTTD of CP was lower for soybean meal (coefficient of variation of 3.2%) and other protein concentrates (sunflower meal, gluten meal, and extruded soybean; coefficient of variation of 4.6% on average), with substitution levels of 30% in all cases [23], than for wheat bran (coefficient of variation of 14.0%) and other cereal by-products (corn gluten feed and DDGs), with a substitution level of 50% in all cases [24], or cereals (a coefficient of variation of 12% for barley, with an inclusion level of 60%; a coefficient of variation of 37.6% for corn, with an inclusion level of 30%) [25].

The precision of the estimates of the aTTD of the fibre of the test feedstuffs was very low, both for aNDFom and ADFom, in particular, even in cases where the contribution of the feedstuff to the fibre content of its respective test diet was remarkable. This is the case of wheat bran, which provided 52.3% of the aNDFom and 33.6% of the ADFom of its test diet, and yet the coefficients of variation were 22.9 and 77.5%, respectively. This fact has also been observed in various studies on rabbits, where the coefficient of variation ranged between 49 and 382% in the aTTD of aNDFom of feedstuffs that contributed between 37 and 50% to the aNDFom of their respective test diets [26,27], or between 29 and 294% in the aTTD of ADFom of feedstuffs that contributed between 6 and 57% to the ADFom of their respective test diets [23,24,25,28]. Compared with the aTTD of GE, OM, and CP, this lack of precision in the estimates of the aTTD of aNDFom and ADFom of the feedstuffs is largely explained by the greater variability in their analytical determination (as evidenced by the greatest discrepancy between the analytical values of the aNDFom and ADFom contents of the test diets and those calculated from the analyses of the corresponding basal diets and test feedstuffs, in comparison with the rest of the analysed nutrients), and by the greater residual variability in their digestibility values in the experimental diets. This can result in values that have no biological meaning, such as the aTTD of aNDFom and ADFom greater than 100% in the case of soybean meal (where the residual variability in their values in the corresponding basal diet was maximum, with a coefficient of variation of 15 and 21%, respectively).

To our knowledge, there is little prior information on the nutritive value for guinea pigs of the feedstuffs evaluated in the current work.

Castro-Bedriñana and Chirinos-Peinado [11] evaluated corn and barley by the substitution method (substitution level of 10%), without reporting the standard errors or the number of animals used. They obtained DE contents (calculated by multiplying the % TDN by 44.09) very similar to those obtained in the current work (3855 vs. 3857 kcal/kg DM for corn; 3518 vs. 3454 kcal/kg DM for barley), although the aTTD of OM was lower (83.3 vs. 90.0% for corn, which, in the aforementioned work, had an anomalously high ash content, 8.4% on DM basis; 80.3 vs. 84.0% for barley). Compared with the current work, the aTTD of CP obtained was higher in the case of corn (87.6 vs. 73.8%) and more similar in the case of barley (63.7 vs. 69.5%).

In the same work, wheat bran was also evaluated by the direct method, without reporting the standard errors or the number of animals used. The obtained DE content (also calculated from % TDN) was also very similar to that found in the current work (2879 vs. 2911 kcal/kg DM). Hidalgo and Valerio [29] also evaluated wheat bran by the direct method, using five animals, and found a DE content of 2801 kcal/kg DM, with greater precision than that obtained in the current work by the substitution method (coefficient of variation of 6 vs. 10%), as expected [21]. Wheat bran is a feedstuff with high variability in its composition, whose DE content for rabbits is mainly inversely related to the ADFom content [30], but comparisons cannot be made because, in the work by Hidalgo and Valerio [29], crude fibre was analysed instead of ADFom.

No other studies have been found on the nutritive value of protein concentrates evaluated in the current work for guinea pigs. In the case of soybean meal, both the DE content and the aTTD of CP were similar to those assigned to soybean meals of similar quality (53.4% CP on average) for rabbits in commonly used databases (3970 kcal/kg DM and 83.9% on average) [10,31]. In the case of the pigeon pea, a lower DE content and higher aTTD of CP were reported for pigs than in the current work for guinea pigs (3421 vs. 3105 kcal/kg DM; 88.4 vs. 61.1%) [32]. Finally, in the case of Leucaena leaf meal, a lower DE content and aTTD of CP were reported for pigs than in the current work for guinea pigs (2679 vs. 2972 kcal/kg DM; 44.0 vs. 68.0%) [33].

## 5. Conclusions

The present work provides the digestible energy content and crude protein digestibility of corn, barley, wheat bran, soybean meal, pigeon pea, and Leucaena leaf meal in guinea pigs (as well as the precision of these values). Further efforts are needed to increase knowledge of the nutritive value of these and other feedstuffs for guinea pigs.

## Figures and Tables

**Table 1 animals-14-03142-t001:** Analytical composition of the evaluated feedstuffs (% on a dry matter basis, unless another unit is indicated).

	Corn	Barley	Wheat Bran	Soybean Meal	Pigeon Pea	Leucaena Leaf Meal
Dry matter (%)	89.9	89.7	89.1	92.3	90.5	91.8
Gross energy (kcal/kg DM)	4331	4251	4534	4590	4376	4738
Ash	1.46	3.12	6.13	7.88	5.11	8.90
Crude protein	8.68	10.0	20.2	53.4	24.2	30.4
Ether extract	2.78	2.40	4.65	2.15	1.39	3.52
Starch	66.7	59.1	22.5	1.42	33.6	1.89
Neutral detergent fibre	14.5	22.8	42.6	10.8	18.1	27.6
Acid detergent fibre	2.41	5.87	11.9	4.41	7.74	12.9
Lignin (sa)	0	0	1.98	0	0	3.70
Amino acids (% crude protein)						
Lysine	2.64	3.54	4.15	5.45	5.67	4.44
Methionine	1.75	1.82	1.72	2.06	1.03	1.12
Cystine	1.94	2.36	2.40	1.52	1.05	0.83
Methionine + Cystine	3.69	4.18	4.13	3.58	2.08	1.95
Threonine	3.16	3.23	3.20	3.43	3.06	3.20
Arginine	3.81	4.26	6.62	6.12	5.03	3.88
Isoleucine	2.69	3.28	3.10	3.88	2.84	2.99
Valine	4.53	4.74	4.64	4.50	3.91	4.30
Histidine	2.29	1.93	2.54	2.11	2.77	1.43
Aspartic acid	6.46	5.77	7.43	11.2	9.06	10.5
Serine	4.38	4.11	4.32	4.73	4.34	3.81
Glutamic acid	16.6	21.6	19.1	17.4	18.9	8.63
Glycine	3.96	4.20	5.35	4.26	3.63	4.13
Alanine	6.23	3.95	4.82	3.83	3.67	4.12
Proline	7.48	9.99	6.14	4.50	3.99	3.65
Tyrosine	1.67	2.46	2.64	3.07	1.88	2.66
Leucine	10.1	6.35	5.92	6.92	5.89	5.87
Phenylalanine	3.35	4.48	3.70	3.77	7.12	3.14

**Table 2 animals-14-03142-t002:** Ingredients (%) and analytical composition (% on a dry matter basis, unless another unit is indicated) of the experimental diets.

	Basal Diet-I	Corn Diet	Barley Diet	Wheat Bran Diet	Basal Diet-II	Soybean Meal diet	Basal Diet-III	Pigeon Pea Diet	Leucaena Leaf Meal Diet
Ingredients									
BM-I ^a^	96.5	56.5	46.5	51.5					
BM-II ^b^					96.5	61.5			
BM-III ^c^							96.5	66.5	66.5
Test feedstuffs		40	50	45		35		30	30
Mineral–vitamin premix	3.5 ^d^	3.5 ^d^	3.5 ^d^	3.5 ^d^	3.5 ^e^	3.5 ^e^	3.5 ^e^	3.5 ^e^	3.5 ^e^
Analytical composition									
Dry matter (%)	92.2	91.5	91.4	91.9	92.0	91.4	91.0	91.3	91.5
Gross energy (kcal/kg DM)	4401	4378	4267	4394	4199	4252	4155	4199	4345
Ash	10.8	7.78	8.12	9.63	7.76	8.84	7.62	8.17	8.75
Crude protein	25.1	19.0	17.7	22.1	14.1	28.3	12.2	16.8	17.7
Neutral detergent fibre	33.5	24.0	26.0	36.2	27.2	22.1	33.4	26.2	30.3
Acid detergent fibre	19.0	11.3	11.4	15.7	12.5	10.1	15.4	12.4	14.3
Lignin (sa)	2.85	1.05	1.03	2.14	1.42	0.75	1.98	1.12	2.40

BM: basal mixture. ^a^ Alfalfa (41.5%), soybean meal (31.1%), barley hulls (23.7%), soybean oil (3.1%), and cane molasses (0.6%). ^b^ Barley (36.3%), alfalfa (21.4%), barley hulls (16.1%), wheat bran (15.5%), corn (9.6%), cane molasses (0.7%), L-arginine (0.25%), and L-lysine HCl (0.15%), first batches. ^c^ Barley (36.3%), alfalfa (21.4%), barley hulls (16.1%), wheat bran (15.5%), corn (9.6%), cane molasses (0.7%), L-arginine (0.25%), and L-lysine HCl (0.15%), second batches. ^d^ Dicalcium phosphate (2.65%), sodium chloride (0.22%), sodium bicarbonate (0.13%), trace element and vitamin premix (0.50%). ^e^ Dicalcium phosphate (2.60%), sodium chloride (0.20%), sodium bicarbonate (0.20%), trace element and vitamin premix (0.50%).

**Table 3 animals-14-03142-t003:** Deviation of the analytical composition of the test diets from their calculated composition (% of the calculated composition).

	Corn Diet	Barley Diet	Wheat Bran Diet	Soybean Meal Diet	Pigeon Pea Diet	Leucaena Leaf Meal Diet
Gross energy	1.8	0.7	0.3	−0.6	0.7	1.4
Organic matter	1.0	0.8	0.9	0.4	0.0	0.6
Crude protein	4.2	3.8	−1.4	3.0	6.7	−0.1
Neutral detergent fibre	−5.5	−5.3	−1.9	4.2	−7.6	−2.9
Acid detergent fibre	−6.7	−5.9	1.6	5.9	−3.2	−0.7

**Table 4 animals-14-03142-t004:** Corrected substitution level and contribution of the test feedstuffs to the analytical composition of their respective test diets (%).

	Corn Diet	Barley Diet	Wheat Bran Diet	Soybean Meal Diet	Pigeon Pea Diet	Leucaena Leaf Meal Diet
Corrected substitution level	41.0	51.6	46.3	35.9	32.1	32.3
Gross energy	39.0	49.4	45.9	37.3	32.1	33.8
Organic matter	42.1	52.3	46.2	34.9	31.9	30.9
Crude protein	18.0	27.9	40.5	65.1	44.4	53.4
Neutral detergent fibre	23.8	43.4	52.3	17.0	21.3	28.2
Acid detergent fibre	8.4	25.6	33.6	15.1	19.2	27.9

**Table 5 animals-14-03142-t005:** Apparent total tract apparent digestibility of the experimental diets (%, unless another unit is indicated; standard error between brackets).

	Basal Diet-I	Corn Diet	Barley Diet	Wheat Bran Diet	Basal Diet-II	Soybean Meal Diet	Basal Diet-III	Pigeon Pea Diet	Leucaena Leaf Meal Diet
Feed intake (g DM/d)	55.7 (2.4)	45.0 (1.6)	41.3 (2.0)	50.2 (1.8)	48.3 (2.5)	58.7 (2.1)	54.7 (3.1)	50.7 (1.2)	43.4 (2.7)
Dry matter	68.7 (0.5)	76.3 (0.6)	75.0 (0.7)	65.6 (1.1)	74.1 (0.8)	77.8 (0.3)	68.7 (0.4)	70.3 (0.4)	67.2 (0.4)
Gross energy	69.3 (0.4)	75.8 (0.7)	74.7 (0.8)	66.7 (1.1)	73.7 (0.8)	78.0 (0.4)	68.6 (0.5)	68.9 (0.5)	65.6 (0.4)
Organic matter	69.3 (0.3)	77.8 (0.6)	76.5 (0.7)	67.0 (1.0)	75.4 (0.8)	79.5 (0.3)	70.3 (0.3)	71.5 (0.4)	68.1 (0.4)
Crude protein	79.6 (0.7)	75.3 (0.7)	73.8 (1.2)	79.4 (1.4)	70.3 (0.9)	80.0 (0.6)	71.0 (0.9)	62.6 (0.7)	69.5 (0.9)
Neutral detergent fibre	47.1 (0.8)	49.9 (1.7)	47.5 (1.4)	44.3 (1.8)	42.6 (2.3)	52.4 (1.0)	39.6 (0.6)	44.0 (0.5)	40.4 (1.1)
Acid detergent fibre	46.7 (0.8)	47.9 (2.0)	39.2 (1.6)	37.4 (2.0)	35.8 (2.8)	46.1 (1.0)	32.6 (0.8)	39.4 (1.3)	31.0 (1.3)

**Table 6 animals-14-03142-t006:** Digestible energy content and apparent total tract digestibility of the test feedstuffs (%, unless another unit is indicated; standard error between brackets).

	Corn	Barley	Wheat Bran	Soybean Meal	Pigeon Pea	Leucaena Leaf Meal
Digestible energy (kcal/kg DM)	3857 (83)	3454 (68)	2911 (110)	3855 (81)	3105 (79)	2972 (72)
Gross energy digestibility	89.1 (1.9)	81.2 (1.6)	64.2 (2.4)	84.0 (1.8)	70.9 (1.8)	62.7 (1.5)
Organic matter digestibility	90.9 (1.4)	84.0 (1.4)	65.6 (2.2)	87.6 (1.7)	73.5 (1.5)	64.1 (1.4)
Crude protein digestibility	73.8 (4.9)	69.5 (4.6)	76.4 (3.6)	88.4 (1.0)	62.1 (1.9)	68.0 (1.8)
Neutral detergent fibre digestibility	47.3 (7.8)	42.0 (3.5)	40.0 (3.5)	110 (12)	45.1 (3.4)	38.4 (4.1)
Acid detergent fibre digestibility	22.1 (25.1)	6.0 (6.8)	21.2 (6.2)	117 (16)	62.6 (7.5)	26.3 (5.2)

## Data Availability

Data are contained within this article. The datasets of the current study are available from the corresponding author upon reasonable request.

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
