# Peer review of "Nutritive Value of Some Concentrate Feedstuffs for Guinea Pigs (Cavia porcellus)"

_animals, 2024, doi:10.3390/ani14213142_

Round 1
Reviewer 1 Report
Comments and Suggestions for Authors
In the manuscript, the authors studied the nutritive value of some concentrate feedstuffs for guinea pigs (Cavia porcellus).
However, the following comments can be made.
1. Materials and methods, paragraph 2.4. Change the title. I suggest - Calculation and statistical processing of data.
2. Conclusion. Expand. Add research results.
Comment for Ethical Approval Concern:
In response to your question on the ethical aspect of the manuscript, I would like to inform you of the following.
The authors have made additions to the manuscript. However, I suggest correcting the text on lines 120-124 as follows.
"A proper ethical assessment of the trials was not performed (due to the lack of an ethics committee for animal experimentation in the UNSAAC). Nevertheless, the experimental protocols were drawn up by one of the authors (Enrique Blas; UPV) in accordance with Spanish legislation on this matter.".
This addition is sufficient to meet the ethical requirements.
I recommend that the manuscript be reviewed for possible approval for acceptance.
Author Response
Dear reviewer,
Thank you for your comments. Please see the attachment.
Kind regards,
Gilbert Alagón

Reviewer 2 Report
Comments and Suggestions for Authors
GENERAL COMMENT:
I consider this work is within the scope of “Animals”. It contains information useful in a field in which available information on the evaluation of feedstuffs for Guinea pigs is scarce. However, some improvement is advisable. I indicate below specific points to be improved in the manuscript.
TITLE:
Type “Cavia porcellus” in italics.
SIMPLE SUMMARY:
It is OK.
ABSTRACT:
Line 31: type “ad libitum” in italics.
KEYWORDS:
These are OK.
INTRODUCTION:
Specific improvement is needed:
Lines 64-65: according to the usage of the journal for bibliographic quotations within the text, replace “de Blas et al. (2019) [10].” with “de Blas et al. [10].”
Lines 66-67: in the same vein, replace “Chirinos-Peinado (2021) [11]” with “Chirinos-Peinado [11]”
Lines 72-73: in the same vein, replace “Crampton et al. (1957) [12]” with “Crampton et al. [12]”
MATERIALS AND METHODS:
Line 86: insert a paragraph break after Table 1 to prevent the paragraph following the table from appearing as a footnote to the table.
Line 110: add “(Peru)” after “Cusco”, thus resulting as: “Cusco (Peru)”.
Line 111: please indicate average (or range of) ambient temperature and humidity in the experimental facility.
Line 113: I recommend adding a bibliographic quotation describing the “Peru” line of Guinea pigs used, in order for the international readers potentially interested to learn more about it.
Line 120: type “ad libitum in italics”.
Line 124: Add “(Royal Decree 53/2013)” following “Spanish legislation on this matter”, thus resulting as: “Spanish legislation on this matter (Royal Decree 53/2013)”.
And the bibliographic reference, to be added to the references section, is:
Ministerio de la Presidencia. Real Decreto 53/2013, de 1 de febrero, por el que se establecen las normas básicas aplicables para la protección de los animales utilizados en experimentación y otros fines científicos, incluyendo la docencia. Boletín Oficial del Estado, 34, 11370-11421.
Line 135: add “(Spain)” after “Universitat Politècnica de València”, thus resulting as: “Universitat Politècnica de València (Spain)”
Lines 140: according to the usage of the journal for bibliographic quotations within the text, replace “AOAC (2002) [14]” with “AOAC [14]”.
Lines 143: in the same vein, replace “Batey (1982) [15],” with “Batey [15],”.
Idem at Lines 149, 151, 152, 158, 168.
RESULTS SECTION:
Line 195: insert a paragraph break after Table 3 to prevent the paragraph following the table from appearing as a footnote to the table.
Line 204: insert a paragraph break after Table 4 to prevent the paragraph following the table from appearing as a footnote to the table.
Lines 224 and 234: “aTTAD” or “aTTD”?
DISCUSSION SECTION:
Overall, this section is OK. However, improvements are needed.
Line 315: according to the usage of the journal for bibliographic quotations within the text, replace “Castro-Bedriñana and Chirinos-Peinado (2021) [11]” with “Castro-Bedriñana and Chirinos-Peinado (2021) [11]”.
Idem at Line 327.
CONCLUSIONS:
Please indicate explicitly the names of the six feedstuffd.
INSTITUTIONAL REVIEW BOARD STATEMENT:
Line 362: Add “(Royal Decree 53/2013)” following “Spanish legislation on this matter”, thus resulting as: “Spanish legislation on this matter (Royal Decree 53/2013)”.
REFERENCES SECTION:
In general terms, this section has a good adjustment to the style and format of the journal for references. However, I recommend reviewing it for removing typos and correct potential flaws. For example:
Type Latin names of the organisms in italics at Lines 427, 432 and 434.
For article titles of articles in languages different to English, please see instructions for authors in order to decide whether these must be in the original language or translated into English.
Etc.
TABLES:
Table 2: Indicate in the footnote the meaning of BM initials.
Author Response
Dear reviewer,
Thank you for your reviews and comments. Please see the attachment.
Kind regards,
Gilbert Alagón

Reviewer 3 Report
Comments and Suggestions for Authors
The paper provides strong justification for the study, and the methodology is sound.
I have only a few comments regarding the results section:
- Lines 63-64: The sentence, "The dietary DE contents were calculated from the DE contents for rabbits of the different feedstuffs..." is unclear. Please rephrase for better clarity.
- In the Materials and Methods section, the software used to calculate the mean, standard error, and coefficient of variation (CV) is not mentioned. CV values are reported in the results, but the methodology section lacks this detail. Please include it.
- There seems to be some inconsistency in the use of terms like "feedstuffs," "experimental diet," "test diet," and "basal diet." While definitions exist, the usage needs to be more consistent throughout the paper.
- The sources of feedstuffs such as corn, barley, and wheat are not specified. Additionally, details on the sampling process are missing. Was the sampling method representative? Please clarify this in the methodology section.
- Lines 147-148: The sentence, "In the case of the pigeon pea and pigeon pea diet, a preliminary..." is a bit unclear. Please revise for better understanding.
- The separate use of terms like "soya bean meal" and "soya bean diet" (or "soya bean meal diet") is confusing. It would be helpful to use these terms consistently throughout the paper.
- Line 174: The phrase "did not provide MO nor..." is unclear. Please clarify the meaning of "MO nor."
Comments on the Quality of English Language
No significant problem in seen except few typographical errors.
Author Response

(The authors gave the same response as above.)

Reviewer 4 Report
Comments and Suggestions for Authors
Dear Authors,
The manuscript presented for evaluation is interesting but should be treated as very preliminary research with increasing importance in the future, especially in certain geographical areas where the mentioned species constitutes a component of the human diet. The experimental design was planned correctly, the parameters of the digestible energy content and crude protein digestibility of six concentrate feedstuffs were assessed. But in my opinion there is a lack of connection between the assumption of the experiment and the future goal (research direction). The authors mention that on the one hand these are laboratory animals, on the other they can be used in the human diet. I miss the connection between the purpose of the described research and the directions of management of these animals and the reference to further research. Do the authors have any additional laboratory markings?
Best regards,
Comment for Ethical Approval Concern:
I don't have comments to this issue.
Author Response

(The authors gave the same response as above.)
